# Emotional Functioning in Long-Term Breast Cancer Survivors: A Cross-Sectional Study on Its Influence and Key Predictors

**DOI:** 10.3390/cancers17091574

**Published:** 2025-05-06

**Authors:** Francisco Álvarez-Salvago, Sandra Atienzar-Aroca, Clara Pujol-Fuentes, Maria Figueroa-Mayordomo, Cristina Molina-García, Palmira Gutiérrez-García, Jose Medina-Luque

**Affiliations:** 1FIBIO Research Group, Department of Physiotherapy, Faculty of Health Sciences, European University of Valencia, 46010 Valencia, Spain; francisco.alvarez2@universidadeuropea.es (F.Á.-S.); clara.pujol@universidadeuropea.es (C.P.-F.); maria.figueroa@universidadeuropea.es (M.F.-M.); gutierrezgarciapalmira@gmail.com (P.G.-G.); jose.med.luque@gmail.com (J.M.-L.); 2Department of Health Sciences, Faculty of Health Sciences, University of Jaén, 23071 Jaén, Spain; 3Department of Dentistry, Faculty of Health Sciences, European University of Valencia, 46010 Valencia, Spain; 4Faculty of Physiotherapy, Podiatry and Occupational Therapy, Catholic University San Antonio-UCAM, 30107 Murcia, Spain; cmolina799@ucam.edu

**Keywords:** emotional functioning, long-term survivorship, breast cancer, health-related quality of life, psychological distress

## Abstract

Breast cancer (BC) survivors often face emotional challenges that can impact their overall well-being, even years after completing treatment. This study explores the factors that influence emotional functioning in long-term breast cancer survivors (LTBCSs) to better understand what contributes to psychological distress or well-being. We analyzed various aspects of health, including health-related quality of life (HRQoL), physical activity, pain, cancer-related fatigue, and mood, to identify the key predictors of emotional health. Our findings show that role functioning, cognitive abilities, self-perceived physical fitness, and mood state play a significant role in emotional well-being. These results highlight the importance of integrating psychological and physical health assessments into follow-up care for BC survivors. By identifying individuals at risk of emotional distress early on, healthcare professionals can design more effective interventions to enhance long-term well-being and HRQoL in this population.

## 1. Introduction

Emotional functioning is a key component of overall well-being, encompassing an individual’s ability to regulate emotions, manage stress, and maintain psychological balance [1]. In cancer patients, particularly long-term breast cancer survivors (LTBCSs) (i.e., ≥5 years post-BC diagnosis), emotional functioning can be significantly affected by persistent physical and psychological challenges [2]. Breast cancer (BC) is one of the most prevalent cancers worldwide among women. The increasing number of BC survivors has led to its recognition as a chronic condition, given that its long-term effects can persist for many years or even throughout a patient’s life [3,4]. Moreover, the treatments required to control the tumor often contribute to a broad spectrum of adverse effects, further impacting the emotional well-being of BC patients and LTBCSs [5]. One of the most relevant manifestations of impaired emotional functioning is psychological distress, an umbrella term encompassing a multifaceted condition characterized by anxiety, depression, mental pain, stress, or discomfort, among other symptoms [6]. In recent years, its impact on health-related quality of life (HRQoL) in BC patients has gained increasing attention, as these individuals experience a high incidence both during and after the disease [7].

If left unaddressed, these emotional changes can interfere with survivors’ overall well-being and complicate their long-term adaptation and recovery process [8]. Additionally, conditions such as depression and anxiety have been suggested as potential factors influencing BC recurrence and survival rates [9]. Comparisons across studies are often challenging due to methodological heterogeneity and the tendency to examine a limited set of variables. Moreover, little research has simultaneously analyzed the factors contributing to the persistence of psychological distress in this population. Among the studies that have explored potential predictors, findings remain inconclusive, with different investigations identifying factors such as comorbidities, parenthood, cognitive status, age, financial difficulties, and overall HRQoL as key determinants [5,9,10,11,12]. Therefore, while studies have explored the impact of emotional state on the health of LTBCSs [5,9,10,11,12], further research is needed to provide a more integrated perspective, allowing for a better understanding of the interplay between emotional functioning, psychological distress and physical and psychological factors. Additionally, identifying consistent predictors could help guide targeted interventions.

Apart from the limited studies on these patients, research highlights the importance of assessing the emotional well-being of LTBCSs [8,10] and the usefulness of tools such as the emotional functioning subscale of the EORTC QLQ-C30, as it addresses this aspect within a broader physical and psychological context [13,14,15,16,17]. Although LTBCSs have completed primary treatment, the persistent fear of recurrence can contribute to difficulties in concentration, as well as increased levels of anxiety or depression [18]. These psychological factors may impact their ability to complete extensive assessments during clinical visits, where time is often limited. In this sense, and despite the well-established role of the emotional functioning subscale in detecting psychological distress [12,13], there is still limited evidence on how LTBCSs’ health status varies based on validated emotional functioning cut-off points. Further research is needed to explore how general health outcomes differ according to emotional functioning levels and to identify the key predictors that influence long-term emotional well-being in this population, which could help create more accurate and personalized interventions.

Therefore, this study aims to explore the possible relationship between emotional functioning on HRQoL, mood state, self-perceived physical fitness, physical activity (PA) level, pain, and cancer-related fatigue (CRF), as well as to identify factors that predict emotional functioning ≥5 years beyond BC diagnosis. Results from this study could support the early detection of at-risk individuals and guide interventions to promote their long-term well-being and HRQoL.

## 2. Materials and Methods

### 2.1. Study Population and Procedures

This study employed a cross-sectional and descriptive design, assessing 80 LTBCSs who were recruited in 2023 through various BC patient and survivor associations in Almería. Sample size calculations were conducted using G*Power (Version 3.1.9.7) for a comparison between two independent groups, determining that a minimum of 72 participants was necessary to achieve a medium effect size (f = 0.25), with a significance level of 0.05 and a statistical power of 0.80. Ultimately, 80 LTBCSs participated. All assessments were carried out within the facilities of the respective BC associations, adhering to the principles of the Declaration of Helsinki (14/2017) [19] and with approval from the Biomedical Research Ethics Committee of Granada (CEIm) (1038-N-16 I.P/07/26/2018).

To be eligible for the study, participants had to be at least 18 years old, have received a diagnosis of stage I–IIIa BC at least five years earlier, and be fluent in Spanish. Women undergoing active cancer treatment, such as chemotherapy, radiotherapy, or hormonal therapy, were not included. Additionally, individuals with diagnosed psychiatric disorders or those taking psychotropic medication were excluded to prevent potential influences on the psychological assessments. Participants who were unable to understand or complete the evaluations independently were also not considered for the study.

As participants were recruited through various BC associations in Almería, the research team first informed these associations about the study, including its objectives, inclusion and exclusion criteria, and assessment procedures. The associations then identified potential participants who met the criteria and informed them about the opportunity to take part in the study. Women interested in participating voluntarily reached out to us, ensuring confidentiality and adherence to data protection regulations.

Once contact was established, participants received further details about the study via phone, and an in-person appointment was scheduled at the association’s facilities. During this meeting, they were provided with a study dossier, and any questions they had were addressed. If they agreed to participate, they signed the informed consent form before undergoing an on-site assessment, which lasted approximately 50 min. To maintain consistency, all evaluations were conducted by the same researcher. Additionally, one team member was responsible for digitalizing the collected data, while another independently analyzed the results.

For group allocation and based on similar recommendations reported in previous studies regarding cut-off points [16], participants were categorized into two groups according to their emotional functioning. The European Organization for Research and Treatment of Cancer Quality of Life Questionnaire Core 30 (EORTC QLQ-C30) Spanish version 3.0 was used, defining the groups as psychological distress (scores ≤ 90) and satisfactory psychological well-being (scores ≥ 91) [16].

### 2.2. Measures

#### 2.2.1. Emotional Functioning

The emotional functioning subscale of the EORTC QLQ-C30 comprises four items that assess affective components related to anxiety, depression, and overall psychological distress. Specifically, it evaluates patients’ self-perceptions of feeling tense, worried, depressed, and irritable [20]. Based on experiences over the previous week, responses are recorded using a four-point Likert scale, ranging from 0 (“not at all”) to 4 (“very much”). The raw scores are then converted into a standardized 0–100 scale, where higher scores reflect better emotional functioning.

For this study, a cut-off score of ≤90 was used to identify psychological distress, while scores ≥91 were categorized as indicative of satisfactory psychological well-being [16]. The Spanish adaptation of the emotional functioning-EORTC-QLQ-C30 has shown strong reliability and validity in the Spanish population and can be completed in under three minutes [21].

#### 2.2.2. Sociodemographic and Clinical Data Collection

Data collection was conducted through structured interviews utilizing a customized questionnaire designed to obtain sociodemographic and clinical information. The clinical variables analyzed included time elapsed since diagnosis, tumor stage, family history of BC, surgical procedures undergone, types of treatment received, current medication use, presence of metastases or recurrence, menopausal status, involvement in psychological or physiotherapy services, and lifestyle factors, such as smoking and alcohol consumption.

#### 2.2.3. Health-Related Quality of Life

HRQoL was assessed using the following two validated instruments: the EORTC QLQ-C30 (version 3.0) and its BC-specific module, the QLQ-BR23. These tools have demonstrated reliability and validity in evaluating HRQoL among cancer patients [21]. Participants provided responses on a four-point Likert scale (ranging from 1 = “not at all” to 4 = “very much”), which were then converted to a standardized 0–100 scale. In the interpretation of scores, higher values on functional and global HRQoL scales indicate better health status; whereas, elevated scores on symptom scales reflect a greater symptom burden. Additionally, a summary score for the QLQ-C30 was calculated by combining 13 scales and individual items, excluding those related to global health status and financial difficulties. In this summary measure, higher scores correspond to improved overall HRQoL [22].

#### 2.2.4. Mood State

Mood state was evaluated using the Scale for Mood Assessment (EVEA), a highly reliable tool with Cronbach’s alpha values between 0.88 and 0.93 [23]. This scale comprises 16 items rated on a 0 to 10 Likert scale, measuring the following four key emotional dimensions: sadness/depression, anxiety, anger/hostility, and happiness. Each dimension’s score is calculated as the mean of its corresponding items, where higher scores reflect a greater intensity of that particular emotional state.

#### 2.2.5. Self-Perceived Physical Fitness

Self-perceived physical fitness was evaluated using the International Fitness Scale (IFIS), a validated questionnaire designed to assess an individual’s perception of their fitness level. This instrument has demonstrated moderate reliability, with a mean weighted Kappa of 0.45 [24]. The IFIS employs a 5-point Likert scale ranging from 1 (very poor) to 5 (very good) and includes five key items measuring overall fitness, as well as specific components, such as cardiorespiratory endurance, muscular strength, speed/agility, and flexibility in comparison to peers. Since it remains relatively stable over time, self-perceived fitness offers meaningful insight into regular PA patterns [25]. Additionally, studies have demonstrated a positive association between self-reported fitness and objectively assessed physical condition, highlighting its validity as an indicator of overall fitness levels [26].

#### 2.2.6. Physical Activity Level

PA level was evaluated using the Minnesota Leisure Time Physical Activity (MLTPA) questionnaire, a validated tool designed to measure the frequency and total duration of PA performed over the previous week. This instrument has demonstrated strong reliability, with an intraclass correlation coefficient (ICC) of 0.95 [27]. The assessment included a structured list of specific activities, and energy expenditure was estimated by multiplying the total weekly hours dedicated to each activity by its respective metabolic equivalent of task (MET) value [28], which represents the energy demand of the activity. Higher values correspond to increased total PA levels. For analysis purposes, PA levels were categorized into the following three groups according to established cut-off points: very low active (≤3 MET), low active (3.1–7.4 MET), and sufficiently active (≥7.5 MET) [29,30,31].

#### 2.2.7. Pain Measures

Pain intensity was assessed using the Visual Analog Scale (VAS), a 10 cm tool with strong reliability (ICC = 0.97) [32], where 0 indicates “no pain”, and 10 represents “worst possible pain”. Participants rated pain in both the affected and unaffected arms. In cases of bilateral BC, the affected arm was determined based on (1) subjective pain perception when comparing the two arms, (2) surgical intervention extent, and (3) the presence of complications, like lymphedema or other post-surgical complications. Additionally, the Brief Pain Inventory (BPI) short form (Cronbach’s α = 0.87–0.89) [33] evaluated pain severity (four items) and its interference in daily life (seven items), with higher scores indicating greater pain burden.

#### 2.2.8. Cancer-Related Fatigue

CRF was measured using the Piper Fatigue Scale (PFS), a 22-item tool designed to assess the following four core dimensions: behavioral impact, emotional response, sensory experience, and cognitive/mood alterations. The total score represents overall CRF intensity, with higher scores indicating greater fatigue severity [34]. This scale has demonstrated strong reliability (Cronbach’s alpha = 0.86) [35]. The following two established classification models were used for CRF severity [34,36]: Model A categorizes CRF as none (0), mild (1–3), moderate (4–6), and severe (7–10); whereas, Model B defines mild CRF as scores between 1–2, moderate as 3–5, and severe as 6–10. Regardless of the classification model used, moderate CRF is considered clinically significant [37]. Patients experiencing moderate-to-severe CRF should be further assessed to rule out underlying conditions requiring medical intervention [38].

### 2.3. Statistical Analysis

The data analysis was conducted using IBM SPSS Statistics for Windows (version 27.0, Armonk, NY, USA). A significance level of *p* < 0.05 was established, with a 95% confidence interval (CI). The Kolmogorov–Smirnov test was employed to examine the normality of all variables, considering a *p*-value > 0.05 as indicative of a normal distribution. To assess group homogeneity in sociodemographic characteristics, t-tests were applied for continuous variables, while Chi-square tests were used for categorical and ordinal variables.

The primary analysis examined emotional functioning, distinguishing between psychological distress and satisfactory psychological well-being as the independent variable. Dependent variables included HRQoL, mood state, self-perceived physical fitness, and CRF (analyzed as a continuous measure). For group comparisons, *t*-tests were used to assess normally distributed continuous variables; whereas, the Mann–Whitney U test was applied for those not following a normal distribution. Results were reported as mean ± standard deviation. Categorical variables, such as CRF and PA levels (classified based on established cut-off values) [29,30,31,34,36], were analyzed using Chi-square tests and presented as percentages. Additionally, effect sizes were determined using Cohen’s *d* and categorized as follows: negligible (*d* = 0–0.19), small (*d* = 0.2–0.49), moderate (*d* = 0.5–0.79), large (*d* = 0.8–1.19), and very large (*d* ≥ 1.20) [39].

Due to the non-normal distribution of most variables, Spearman correlation analysis was conducted to examine the relationship between emotional functioning—measured through the EORTC QLQ-C30 subscale—and other study variables. For this part of the analysis, emotional functioning was analyzed as a continuous dependent variable. Additionally, a stepwise multiple linear regression identified key factors influencing long-term emotional functioning variability. Variables were included in the regression model if they showed a significant correlation with the dependent variable and if inter-variable correlations remained below 0.70 to minimize collinearity [40]. A forward selection approach was applied, adding significant predictors in order of their association strength. At each step, model significance was assessed, and standardized β coefficients were calculated for the final model to allow for a comparison of the relative impact of the independent variables on emotional functioning.

## 3. Results

### 3.1. Demographic and Clinical Characteristics

Participants were categorized as reporting psychological distress (47.50%) and satisfactory psychological well-being (52.50%) [16]. No significant differences were found between groups in terms of the demographic and clinical characteristics of the 80 LTBCSs.

The mean age for participants with psychological distress was 49.76 ± 7.20 years, while for those with a satisfactory psychological well-being, it was 49.04 ± 8.83 years. In the group with psychological distress, 15.8% were divorced or separated, 39% were on sick leave, 84.2% had undergone both radiotherapy and chemotherapy, and 55.3% had attended sessions with a psychologist in the last 3 months. With regards to LTBCSs with satisfactory psychological well-being, 4.8% were divorced or separated, 35.7% were on sick leave, 92.9% had undergone both radiotherapy and chemotherapy, and 61.9% had attended sessions with a psychologist in the last 3 months. Further information on the demographic and clinical characteristics is presented in Table 1.

### 3.2. Health-Related Quality of Life

The analysis of HRQoL, assessed using the QLQ-C30, revealed significant differences between groups. LTBCSs experiencing psychological distress had significantly lower scores for “global health status”, “summary score”, and all functioning scales except for “physical functioning”. They also reported higher levels of “fatigue”, “nausea and vomiting”, “pain”, “dyspnea”, “insomnia”, “appetite loss”, and “financial difficulties” compared to LTBCSs with satisfactory psychological well-being (U = 247.00 to 701.00; *p* < 0.01 to 0.03; *d* = 0.36 to >1.20). No significant differences were found in the remaining comparisons between the groups (*p* > 0.05). See Table 2 for further details.

Additionally, significant differences between the groups were also observed in the BR23 module. LTBCSs with psychological distress had significantly lower scores for “body image” and “sexual functioning”, as well as higher scores for “systemic therapy side effects”, “breast symptoms”, “arm symptoms”, and “upset by hair loss” compared to the other group (U = 305.50 to 743.50; *p* < 0.01 to 0.01; *d* = 0.58 to 1.19) (refer to Table 2).

### 3.3. Mood State

The analysis of mood state, assessed using the EVEA, revealed significant differences across all EVEA domains between groups. LTBCSs experiencing psychological distress reported higher levels of “sadness–depression”, “anxiety”, and “anger–hostility”, as well as lower levels of “happiness” compared to LTBCSs with satisfactory psychological well-being (U = 475.50 to 511.10; all *p* < 0.01; *d* = 0.06 to 0.70). Table 3 provides a visual summary of these results.

### 3.4. Self-Perceived Physical Fitness

The analysis of physical fitness, assessed using the IFIS, revealed significant differences between groups across all IFIS domains. LTBCSs experiencing psychological distress reported lower levels of self-perceived “general physical fitness”, “cardiorespiratory endurance”, “muscular strength”, “speed-agility”, and “flexibility” compared to LTBCSs with satisfactory psychological well-being (U = 290.00 to 475.00; all *p* < 0.01; *d* = 0.79 to >1.20) (Table 3).

### 3.5. Physical Activity Level

The analysis of between-group differences in MLTPA scores revealed significant differences (*p* < 0.01). Specifically, LTBCSs experiencing psychological distress reported higher levels of inactivity (31.6%) compared to the other group (21.4%). Interestingly, the analysis also showed that 42.9% of LTBCSs with satisfactory psychological well-being met the minimum PA activity recommendations [29,30,31], compared to only 23.7% of LTBCSs experiencing psychological distress (Table 3).

### 3.6. Pain

The analysis of pain revealed significant between-group differences in both the VAS and BPI measures. In this regard, LTBCSs experiencing psychological distress reported higher pain levels in both the affected and non-affected arms (VAS), as well as greater pain intensity and interference (BPI) compared to those with satisfactory psychological well-being (U = 444.00 to 561.00; *p* < 0.01 to 0.01; *d* = 0.49 to 0.86) (Table 3).

### 3.7. Cancer-Related Fatigue

The analysis of PFS domains revealed significant differences between the groups. LTBCSs experiencing psychological distress reported higher levels of CRF across all domains compared to those with satisfactory psychological well-being (U = 360.50 to 416.50; all *p* < 0.01; *d* = 0.85 to 1.05). Regarding the two cut-scores [34,36], A and B, the analysis showed that LTBCSs experiencing psychological distress had significantly higher rates of “moderate” to “severe” CRF for both cut-scores (A = 65.8% and B = 76.3%) compared to those with satisfactory psychological well-being (A = 19% and B = 21.4%) (both *p* < 0.01) (Table 3).

### 3.8. Correlation Analysis

The Spearman’s correlation analysis revealed significant positive correlations between the level of emotional functioning and the following variables: QLQ-C30: “role functioning”, “cognitive functioning”, “social functioning”, and “fatigue”, QLQ-BR23: “body image”, “sexual functioning”, and “systemic therapy side effects”, EVEA: “happiness”, IFIS: “general physical fitness”, “cardiorespiratory endurance”, “muscular strength”, “speed/agility”, and “flexibility”, MLTPA: “PA level” (ρ = 0.271 to 0.744; *p* < 0.01 to 0.01). While significant negative correlations were observed between the level of emotional functioning and the following variables: “family history of BC”, PFS: “behavioral/severity”, “affective”, “sensory”, “cognitive”, and “total fatigue score”, QLQ-C30: “nausea and vomiting”, “pain”, “dyspnea”, “insomnia”, “appetite loss”, “constipation”, “diarrhea”, “financial difficulties”, and “global health”, QLQ-BR23: “breast symptoms”, “arm symptoms”, and “upset by hair loss”, VAS: “affected arm” and “non-affected arm”, BPI: “intensity” and “interference”, EVEA: “sadness–depression”, “anxiety”, and “anger–hostility” (ρ = −0.277 to −0.657; *p* < 0.01 to 0.03). Refer to Figure 1 for additional details.

### 3.9. Multiple Linear Regression Analysis

The final regression model identified the following variables as significant predictors of higher levels of emotional functioning: “role functioning” and “cognitive functioning” from the QLQ-C30, as well as “self-perceived general physical fitness” from the IFIS and “sadness–depression” from the EVEA. Together, these factors explained 64.2% of the variance in emotional functioning (*r^2^* adjusted = 0.642; *p* < 0.01 to 0.04) in individuals who were ≥5 years post-cancer diagnosis. The results are presented in Table 4.

## 4. Discussion

The primary objective of this study was to examine the association between emotional functioning and overall health in LTBCSs. Additionally, it aimed to identify key predictors of emotional functioning in this population at least five years post-diagnosis. The findings revealed that 47.50% of LTBCSs experienced psychological distress, while 52.50% maintained satisfactory emotional well-being. Those with psychological distress showed lower HRQoL, poorer mood, and diminished self-perceived physical fitness, along with greater physical inactivity, pain, and CRF. Moreover, the combination of “role functioning”, “cognitive functioning”, “self-perceived physical fitness”, and “sadness–depression” explained 64.2% of the variance in emotional functioning among LTBCSs.

With regards to HRQoL, those experiencing psychological distress exhibited a higher symptom burden, poorer functioning, and diminished overall HRQoL, as assessed by the QLQ-C30 and BR23 instruments. Emotional functioning plays a crucial role in overall well-being, influencing the ability to regulate emotions, cope with stress, and maintain psychological balance [1]. Given that psychological distress encompasses a broad range of emotional difficulties—including anxiety, depression, and stress—LTBCSs with higher distress levels demonstrated significantly worse health HRQoL outcomes, which is consistent with previous findings in this field [41,42]. These impairments were reflected not only in increased CRF, pain, and insomnia but also in challenges related to body image perception and sexual functioning, further complicating long-term adaptation to survivorship [11,43].

Despite the growing recognition of psychological distress in BC survivors, a major gap remains in understanding its long-term implications beyond five years post-diagnosis. Most studies focus on patients undergoing active treatment or in the early survivorship phase, often overlooking the persistent emotional burden that many LTBCSs continue to experience [7,44]. Additionally, while anxiety and depression are widely studied, fewer investigations consider psychological distress as a broader construct that encompasses multiple emotional challenges, including cognitive and social dimensions. This limitation highlights the need for further research exploring the interplay between emotional functioning and HRQoL in LTBCSs, particularly in identifying the predictors of distress that could inform targeted interventions. Developing strategies to systematically assess and address emotional functioning in this population is essential for improving long-term adaptation and overall HRQoL.

As for mood state, those experiencing psychological distress reported higher levels of “sadness–depression”, “anxiety”, and “anger–hostility”, as well as lower levels of “happiness” compared to LTBCSs with satisfactory psychological well-being. These findings are consistent with previous research indicating that BC survivors are at an increased risk of experiencing symptoms of depression and anxiety compared to the general population, even up to 10 years post-diagnosis [42]. Moreover, a systematic review and meta-analysis have highlighted that poor emotional regulation is associated with greater psychological distress in cancer survivors, emphasizing the importance of addressing these factors in long-term care [45]. Although not statistically significant, the higher prevalence of family history of BC among participants with psychological distress (*p* = 0.06) may reflect a clinically relevant trend. The absence of genetic testing data in our sample limits further interpretation; however, it is plausible that familial cancer experiences or perceived hereditary risk could influence long-term psychological outcomes. Similarly, although marital status did not differ significantly between groups (*p* = 0.42), it is important to note that this variable was also assessed five years after diagnosis and may, therefore, reflect both pre-existing and post-diagnosis relational dynamics. The higher proportion of divorced or separated individuals among those with psychological distress (15.8% vs. 4.8%) could be relevant but must be interpreted with caution. Further research is needed to explore the directionality and potential psychological impact of both family history and marital status LTBCSs. These findings highlight the importance of not only continuing to investigate such sociodemographic and familial factors but also of developing comprehensive psychosocial interventions aimed at enhancing emotional regulation and addressing psychological distress in LTBCSs. Implementing routine mood assessments and providing tailored psychological support could be instrumental in improving their overall well-being and HRQoL.

Considering self-perceived physical fitness, our findings indicate that LTBCSs experiencing psychological distress reported significantly lower self-perceived physical fitness across all domains—general physical fitness, cardiorespiratory endurance, muscular strength, speed–agility, and flexibility—compared to those with satisfactory psychological well-being. While research in adolescents and short-term BC survivors has identified a link between higher self-perceived physical fitness and improved mood or lower psychological distress [46,47,48,49], evidence within long-term survivorship remains limited. Our findings also showed that LTBCSs experiencing psychological distress were more likely to be inactive and less likely to meet minimum PA recommendations [29,30,31] compared to their counterparts with satisfactory psychological well-being. This aligns with previous studies demonstrating that physical exercise significantly reduces depression and anxiety among BC survivors [50]. These observations highlight the intricate relationship between psychological distress, self-perceived physical fitness, and PA levels in LTBCSs. Addressing psychological distress through targeted interventions may enhance self-perceived physical fitness and encourage greater engagement in PA, thereby improving overall health outcomes in this population.

In relation to pain, our results indicate that LTBCSs experiencing psychological distress report higher pain levels in both affected and non-affected arms, as well as greater pain intensity and interference. Chronic pain is a well-documented issue in BC patients and survivors and has been strongly associated with increased psychological distress, including anxiety and depression [7,51]. However, most evidence comes from studies conducted in patients undergoing active treatment or in short-term survivors, while research specifically addressing this relationship in long-term survivorship remains more limited. Some studies suggest that surgical interventions may play a role in the persistence of pain-related distress. More aggressive procedures, such as mastectomy, have been linked to higher levels of psychological distress in LTBCSs [5]. In our study, a higher percentage of LTBCSs in the psychological distress group had undergone unilateral (31.6%) or bilateral (7.9%) mastectomy compared to those with satisfactory psychological well-being (21.4% and 2.4%, respectively), which could partly explain the observed differences in pain perception. Given the bidirectional relationship between pain and psychological distress, further research is needed to clarify whether these associations persist in long-term survivorship and to develop integrated interventions targeting both pain and emotional well-being.

When it comes to CRF, the prevalence of moderate-to-severe CRF was significantly higher in LTBCSs experiencing psychological distress, with 65.8% and 76.3% classified as moderate–severe under cut-score Models A and B [34,36], respectively. These findings align with previous research, indicating that psychological distress, particularly depression and anxiety, can exacerbate CRF severity [52,53]. One plausible explanation for this association is that psychological distress often leads to sleep disturbances, which in turn impair physical recovery and increase the perception of fatigue [54]. However, since we lack data on sleep quality in our LTBCSs, we cannot fully confirm this hypothesis. Similarly, another potential explanation could be that greater emotional deterioration leads to a tendency toward physical inactivity (as most of our LTBCSs experiencing psychological distress do not meet PA recommendations) [29,30,31], which in turn contributes to the persistence or worsening of CRF levels [55]. Future studies should explore targeted interventions aimed at improving emotional well-being, promoting regular exercise, and enhancing sleep quality to better manage CRF and improve long-term survivorship outcomes.

The correlation analysis revealed positive associations between emotional functioning and other functional domains, body image, happiness, and self-perceived physical fitness. In contrast, it was negatively associated with CRF, symptoms like pain, dyspnea, insomnia, appetite loss, constipation, and diarrhea, as well as economic difficulties, depression, anxiety, and anger–hostility. Additionally, the multiple regression analysis identified role functioning and cognitive functioning (QLQ-C30), self-perceived general physical fitness (IFIS), and sadness–depression (EVEA) as significant predictors of emotional functioning, collectively explaining 64.2% of its variance. These results suggest that, beyond the impact of specific symptoms, the ability to engage in daily activities, maintain cognitive abilities, and perceive oneself as physically fit may serve as protective factors for emotional well-being in LTBCSs.

These findings are consistent with prior research, demonstrating the persistent emotional burden in LTBCSs [5,9]. Breidenbach et al. (2022) found that depression and anxiety levels were higher 5 to 6 years post-diagnosis than at 40 weeks, with significant predictors including lower vocational status, having children, and comorbidities [5]. Similarly, Götze et al. (2020) reported that 17% of LTBCSs experienced moderate-to-severe depression, with financial difficulties, impaired cognitive function, and poor global HRQoL as the strongest contributing factors [9]—factors that also showed significant correlations with emotional functioning in our study.

In contrast, Ren et al. (2024) identified older age, recurrence, and advanced stage at diagnosis as predictors of long-term impairments in emotional and functional well-being in LTBCSs [10]. While our results did not indicate an association between emotional functioning and these factors, both studies highlight the complex and multifaceted nature of emotional distress in long-term cancer survivorship. Additionally, De la Torre-Luque et al. (2020) emphasized the role of insomnia and worry in sustaining emotional distress among LTBCSs [12], findings that resonate with our observed correlations between emotional functioning and sleep disturbances.

Despite these contributions, there is still limited consensus on the key predictors of emotional functioning in LTBCSs. Different studies have identified a wide range of predictors, with some aligning with our findings and others differing due to variations in statistical methods, assessment tools, and study designs [5,9,10,12]. This heterogeneity makes direct comparisons challenging and underscores the need for further research to refine and consolidate the understanding of emotional well-being in this population. Achieving greater consensus in predictive factors through standardized methodologies could enhance comparability between studies and improve the design of targeted interventions. Overall, while previous studies have explored various determinants of emotional distress in LTBCSs, our findings highlight the significant predictive role of cognitive and role functioning, self-perceived physical fitness, and sadness–depression. The 64.2% variance explained reinforces the relevance of integrating both functional and psychological assessments in survivorship care, allowing for the early identification of individuals at risk and the development of more personalized interventions.

This study has several limitations that should be acknowledged. The cross-sectional design of this study precludes causal interpretations, and the use of self-reported measures may introduce recall or response bias. However, previous studies support the validity of self-reported assessments in evaluating both physical and psychological well-being, as they can effectively differentiate between individuals with varying levels of functioning [25,26]. Nonetheless, future research should consider incorporating objective assessments to complement self-reported data and provide a more comprehensive evaluation of emotional functioning in LTBCSs. Additionally, the exclusive inclusion of female LTBCSs limits the generalizability of these findings to male BC survivors, who, despite representing approximately 1% of cases, remain under-represented in survivorship research [56]. Another limitation is the lack of pre-diagnosis psychological health data, which prevents determining whether the observed emotional distress is a consequence of cancer survivorship or an exacerbation of pre-existing psychological conditions. Longitudinal studies tracking emotional functioning from diagnosis onwards could help clarify these associations. Finally, and regarding the classification of emotional functioning, we selected a cut-off point of 90, as the findings of the previous study indicated that individuals scoring below this threshold might have unaddressed psychological needs [16]. While, in another investigation, the authors suggest lower cut-off values (ranging from 73.1 to 75.9) for populations with a mean age of 49 years [57] (mean age among our LTBCSs), applying these thresholds in our study could have led to the misclassification of individuals with psychological distress as having good emotional well-being. Given that LTBCSs may experience persistent but unrecognized psychological challenges, we prioritized a more conservative threshold to avoid overlooking unmet psychological needs. Future research should further investigate the appropriateness of different cut-off points in this population, considering variations in age, survivorship stage, and psychological distress presentation.

Despite its limitations, this study provides valuable insights into the determinants of emotional functioning in LTBCSs. While previous research has explored various contributors to emotional distress [5,9,10,12], our findings highlight the predictive role of role functioning, cognitive functioning, self-perceived physical fitness, and sadness–depression—factors that have received comparatively less attention. This perspective expands the understanding of emotional distress in survivorship, reinforcing the relevance of functional and psychological assessments in long-term care. The use of validated instruments tailored to cancer populations strengthens the reliability of our findings. Additionally, the regression model, explaining 64.2% of the variance in emotional functioning, underscores the clinical significance of the identified predictors. By incorporating self-reported assessments—recognized as efficient and strongly correlated with objective measures [25,26]—this study offers a feasible approach for identifying individuals at greater risk of emotional distress. These insights support the development of targeted interventions aimed at enhancing both psychological distress and physical well-being in LTBCSs.

## 5. Conclusions

In conclusion, nearly half of the LTBCSs (47.50%) experienced psychological distress, which was associated with poorer HRQoL, lower mood, reduced self-perceived physical fitness, and higher levels of physical inactivity, pain, and CRF. Furthermore, role functioning, cognitive functioning, self-perceived physical fitness, and sadness–depression emerged as significant predictors of emotional functioning, collectively explaining 64.2% of its variance.

These findings underscore the complex interplay between emotional well-being and key functional and psychological factors in long-term survivorship. From a clinical perspective, routine assessment of emotional functioning, cognitive abilities, and self-perceived physical fitness could enhance early detection of at-risk individuals. Implementing personalized interventions—such as structured exercise programs, psychoeducational strategies, and cognitive–behavioral approaches—may help mitigate emotional distress and improve both physical and psychological well-being in this population. Future research should focus on optimizing multidisciplinary care models to better support LTBCSs in maintaining long-term emotional and functional well-being.

## Figures and Tables

**Figure 1 cancers-17-01574-f001:**
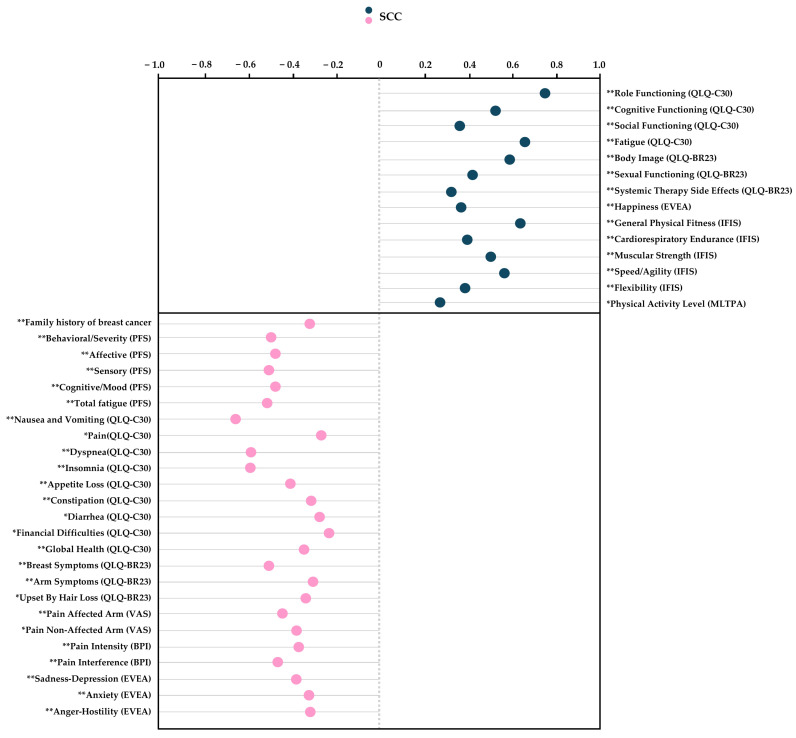
Spearman’s correlation coefficient for emotional functioning using the QLQ-C30. Abbreviations: QLQC30, The EORTC Core Quality of Life Quality of Life Questionnaire; QLQBR23, The Breast Cancer-Specific Module; PFS, Piper Fatigue Scale; VAS, Visual Analogue Scale; IFIS, International Fitness Scale; MLTPA, Minnesota Leisure Time Physical Activity; EVEA, Scale for Mood Assessment; SCC, Spearman’s correlation coefficient. Note: For this analysis, the emotional functioning data as the dependent variable were used in their non-categorized version. * *p* < 0.05; ** *p* < 0.01.

**Table 1 cancers-17-01574-t001:** Demographic, clinical, and medical characteristics of the groups.

Characteristics	LTBCSs Emotional Functioning		
	Psychological Distress	Satisfactory Psychological Well-Being	*p/X* ^2^
	≤90 (QLQ-C30) (*n* = 38)	≥91 (QLQ-C30) (*n* = 42)	
Mean age ± SD, years	49.76 ± 7.20	49.04 ± 8.83	0.57 ^a^
Mean time since diagnosis ± SD, months	90.97 ± 28.56	90.01 ± 30.15	0.64 ^a^
Mean time since the first surgery ± SD, months	87.89 ± 29.04	87.34 ± 30.55	0.69 ^a^
Marital Status, *n* (%)			
Not married	6 (15.8)	7 (16.7)	
Married	24 (63.2)	31 (73.8)	
Divorced or separated	6 (15.8)	2 (4.8)	
Widowed	2 (5.3)	2 (4.8)	0.42 ^b^
Educational level, *n* (%)			
Primary school	18 (47.4)	16 (38.1)	
Secondary school	10 (26.3)	8 (19.0)	
University	10 (26.3)	18 (42.9)	0.29 ^b^
Employment Status, *n* (%)			
Housewife	13 (34.2)	12 (28.6)	
Currently working	5 (13.2)	12 (28.6)	
Sick leave	15 (39.5)	15 (35.7)	
Retired	5 (13.2)	3 (7.1)	0.35 ^b^
Tobacco consumption, *n* (%)			
Non-consumption	18 (47.4)	22 (52.4)	
Smoker	11 (28.9)	8 (19.0)	
Ex-smoker	9 (23.7)	12 (28.6)	0.57 ^b^
Alcohol consumption, *n* (%)			
Non-consumption	14 (36.8)	16 (38.1)	
Monthly	12 (31.6)	7 (16.7)	
Weekly	9 (23.7)	18 (42.9)	
Daily	3 (7.9)	1 (2.4)	0.15 ^b^
Family history of breast cancer, *n* (%)			
No	12 (31.6)	27 (64.3)	
Yes	26 (68.4)	15 (35.7)	0.06 ^b^
Menopause, *n* (%)			
No	4 (10.5)	7 (16.7)	
Yes	34 (89.5)	35 (83.3)	0.42 ^b^
Tumor stage, *n* (%)			
I	7 (18.4)	16 (38.1)	
II	26 (68.4)	20 (47.6)	
III ^a^	5 (13.2)	6 (14.3)	0.12 ^b^
Type of treatment, *n* (%)			
None	0 (0)	0 (0)	
Radiotherapy	3 (7.9)	0 (0)	
Chemotherapy	3 (7.9)	3 (7.1)	
Radiotherapy and chemotherapy	32 (84.2)	39 (92.9)	0.17 ^b^
Surgery, *n* (%)			
Lumpectomy	6 (15.8)	5 (23.8)	
Quadrantectomy	17 (44.7)	17 (52.4)	
Unilateral mastectomy	12 (31.6)	11 (21.4)	
Bilateral mastectomy	3 (7.9)	3 (2.4)	0.41 ^b^
Type of medication, *n* (%)			
None	8 (21.1)	11 (26.2)	
Tamoxifen	14 (36.8)	17 (40.5)	
Other types	16 (42.1)	14 (33.3)	0.70 ^b^
Metastasis, *n* (%)			
No	30 (78.9)	36 (85.7)	
Yes	8 (21.1)	6 (14.3)	0.42 ^b^
Recurrence, *n* (%)			
No	32 (84.2)	35 (83.3)	
Yes	6 (15.8)	7 (16.7)	0.91 ^b^
Currently seeing a psychologist or in the last three months, *n* (%)			
No	17 (44.7)	16 (38.1)	
Yes	21 (55.3)	26 (61.9)	0.54 ^b^
Currently seeing a physiotherapist or in the last three months, *n* (%)			
No	13 (34.2)	16 (38.1)	
Yes	25 (65.8)	26 (61.9)	0.71 ^b^

Abbreviations: LTBCSs, long-term breast cancer survivors; QLQC30, The EORTC Core Quality of Life Quality of Life Questionnaire; *n*, sample size; SD, standard deviation. *p* values for between-group differences were calculated using *t*-test ^a^ and *X*^2^ for categorical variables ^b^.

**Table 2 cancers-17-01574-t002:** Health-related quality of life values between groups.

Variables	LTBCSs Emotional Functioning		*p* Values	Cohen’s *d*
	Psychological Distress	Satisfactory Psychological Well-Being		
	≤90 (QLQ-C30) (*n* = 38)	≥91 (QLQ-C30) (*n* = 42)		
Functioning Scales QLQ-C30, mean ± SD, median; IQR, and (95% CI)				
*Physical Functioning*	35.08 ± 21.84 33.33; 0.00 (27.90–42.26)	29.75 ± 16.26 33.33; 0.00 (24.68–34.82)	0.21	0.28
*Role Functioning*	69.18 ± 21.33 73.33; 26.67 (62.17–76.19)	92.83 ± 10.85 93.33; 12.58 (89.44–96.21)	<0.01 **	>1.20
*Cognitive Functioning*	50.65 ± 30.90 50.00; 45.83 (40.50–60.81)	77.97 ± 24.47 79.16; 33.33 (70.34–85.60)	<0.01 **	0.98
*Social Functioning*	53.06 ± 32.61 50.00; 50.00 (42.34–63.79)	69.84 ± 27.35 83.33; 37.50 (61.31–78.36)	0.01 *	0.56
Symptom Scales QLQ-C30, mean ± SD, median; IQR, and (95% CI)				
*Fatigue*	88.88 ± 21.03 100.00; 16.67 (82.33–95.44)	54.38 ± 32.34 58.33; 37.51 (43.75–65.01)	<0.01 **	>1.20
*Nausea and Vomiting*	54.96 ± 28.05 44.44; 55.55 (45.74–64.19)	19.84 ± 23.98 11.11; 33.33 (12.36–27.31)	<0.01 **	>1.20
*Pain*	10.96 ± 21.32 0.00; 16.67 (3.95–17.97)	4.36 ± 14.28 0.00; 0.00 (−0.08–8.81)	0.03 *	0.36
Single Items QLQ-C30, mean ± SD, median; IQR, and (95% CI)				
*Dyspnea*	55.70 ± 30.82 50.00; 50.00 (45.56–65.82)	23.41 ± 25.25 16.67; 33.33 (15.54–31.28)	<0.01 **	1.15
*Insomnia*	38.59 ± 33.35 33.33; 0.52 (27.63–49.56)	9.52 ± 21.19 0.00; 0.00 (2.91–16.12)	<0.01 **	1.04
*Appetite Loss*	62.28 ± 30.18 66.67; 66.67 (52.35–72.20)	39.28 ± 34.10 33.33; 66.67 (28.65–49.91)	<0.01 **	0.71
*Constipation*	16.66 ± 29.76 0.00; 33.33 (6.88–26.44)	7.14 ± 20.20 0.00; 0.00 (0.84–13.44)	0.06	0.37
*Diarrhea*	32.45 ± 36.75 33.33; 66.67 (20.37–44.53)	18.65 ± 25.55 0.00; 33.33 (10.68–26.61)	0.11	0.44
*Financial Difficulties*	18.42 ± 28.68 0.00; 33.33 (8.99–27.84)	6.34 ± 15.15 0.00; 0.00 (1.62–11.07)	0.03	0.53
Global Health Status QLQ-C30, mean ± SD, median; IQR, and (95% CI)				
*Global Health Status*	14.28 ± 28.64 0.00; 33.33 (5.35–23.21)	32.54 ± 36.67 0.00; 0.00 (20.48–44.59)	<0.01 **	0.56
Summary Score QLQ-C30, mean ± SD, median; IQR, and (95% CI)				
*Summary Score*	56.81 ± 15.32 59.10; 20.07 (51.77–61.85)	73.79 ± 10.59 76.02; 12.10 (70.48–77.09)	<0.01 **	>1.20
Functional Scales QLQ-BR23, mean ± SD (95% CI)				
*Body Image*	51.53 ± 18.96 50.00; 33.34 (45.29–57.77)	74.60 ± 19.90 75.00; 25.00 (68.40–80.80)	<0.01 **	1.19
*Sexual Functioning*	66.22 ± 31.17 75.00; 50.00 (55.98–76.47)	87.10 ± 20.59 100; 16.67 (80.68–93.52)	<0.01 **	0.79
*Sexual Enjoyment*	19.73 ± 23.84 16.67; 33.33 (11.90–27.57)	23.80 ± 18.45 33.33; 33.33 (18.05–29.55)	0.16	0.19
*Future Perspective*	30.70 ± 26.14 33.33; 33.33 (22.10–39.29)	33.33 ± 24.41 33.33; 0.00 (25.72–40.94)	0.55	0.10
Symptom Scales QLQ-BR23, mean ± SD, median; IQR, and (95% CI)				
*Systemic Therapy Side Effects*	65.08 ± 39.61 66.67; 66.67 (52.73–77.42)	42.98 ± 33.69 33.33; 66.67 (31.90–54.05)	<0.01 **	0.60
*Breast Symptoms*	37.84 ± 20.71 33.33; 28.57 (31.03–44.65)	20.26 ± 20.84 14.29; 23.81 (13.76–20.76)	<0.01 **	0.85
*Arm Symptoms*	34.21 ± 30.55 25.00; 47.75 (24.16–44.25)	18.84 ± 22.09 12.50; 25.00 (11.96–25.73)	0.01 *	0.58
*Upset By Hair Loss*	41.81 ± 33.12 38.88; 66.67 (30.92–52.70)	22.75 ± 25.84 22.22; 33.33 (14.69–30.80)	<0.01 **	0.64

Abbreviations: LTBCSs, long-term breast cancer survivors; QLQC30, The EORTC Core Quality of Life Quality of Life Questionnaire; QLQBR23, The Breast Cancer-Specific Module; CI, confidence interval; *n*, sample size; SD, standard deviation; IQR, inter-quartile range. Note: The row corresponding to emotional functioning (functional scales QLQ-C30) has not been included in the table as it is the independent grouping variable. *p* values for between-group differences were calculated using the Mann–Whitney U test. Between-group effect sizes were calculated using Cohen’s *d.* * *p* < 0.05; ** *p* < 0.01.

**Table 3 cancers-17-01574-t003:** Mood state, self-perceived physical fitness, physical activity level, pain, and cancer-related fatigue values between groups.

Variables		LTBCSs Emotional Functioning		*p/X* ^2^	Cohen’s *d*
		Psychological Distress	Satisfactory Psychological Well-Being		
		≤90 (QLQ-C30) (*n* = 38)	≥91 (QLQ-C30) (*n* = 42)		
EVEA, mean ± SD, median; IQR, and (95% CI) ^a^					
*Sadness–depression*		3.86 ± 2.83 4.12; 5.06 (2.93–4.79)	2.06 ± 2.29 1.12; 3.56 (1.34–2.78)	<0.01 **	0.70
*Anxiety*		3.95 ± 2.72 4.25; 4.63 (3.05–4.85)	2.24 ± 2.23 1.50; 2.75 (1.54–2.94)	<0.01 **	0.69
*Anger–hostility*		3.12 ± 2.67 2.62; 4.50 (2.24–4.00)	1.45 ± 2.04 0.50; 2.00 (0.82–2.09)	<0.01 **	0.70
*Happiness*		5.79 ± 8.44 4.12; 2.50 (3.01–8.57)	6.17 ± 2.41 6.62; 3.50 (5.42–6.92)	<0.01 **	0.06
IFIS, mean ± SD, median; IQR, and (95% CI) ^a^					
*General physical fitness*		2.68 ± 0.77 3.00; 1.00 (2.42–2.93)	3.83 ± 0.85 4.00; 2.00 (3.56–4.09)	<0.01 **	>1.20
*Cardiorespiratory endurance*		2.44 ± 1.00 2.50; 1.00 (2.11–2.77)	3.28 ± 1.04 3.00; 1.25 (2.96–3.61)	<0.01 **	0.82
*Muscular strength*		2.42 ± 0.82 2.00; 1.00 (2.14–2.69)	3.28 ± 1.01 3.00; 1.25 (2.96–3.60)	<0.01 **	0.93
*Speed/agility*		2.39 ± 0.71 2.00; 1.00 (2.15–2.63)	3.45 ± 0.88 3.00; 1.00 (3.17–3.72)	<0.01 **	>1.20
*Flexibility*		2.47 ± 0.89 2.50; 1.00 (2.18–2.76)	3.23 ± 1.03 3.00; 1.00 (2.91–3.55)	<0.01 **	0.79
MLTPA, *n* (%) ^b^					
*Inactive: ≤3 (MET hour/week)*		12 (31.6)	9 (21.4)	<0.01 **	
*Low active: 3.1–7.4 (MET hour/week)*		17 (44.7)	15 (35.7)		
*Active: ≥7.5 (MET hour/week)*		9 (23.7)	18 (42.9)		-
VAS (cm), mean ± SD, median; IQR, and (95% CI) ^a^					
*Dominant arm*		3.39 ± 2.79 4.00; 5.25 (2.47–4.31)	1.33 ± 1.94 0.00; 2.25 (0.72–1.93)	<0.01 **	0.86
*Non-dominant arm*		2.13 ± 2.94 0.50; 3.50 (1.16–3.09)	0.85 ± 2.29 0.00; 0.00 (0.14–1.57)	<0.01 **	0.49
BPI, mean ± SD, median; IQR, and (95% CI) ^a^					
*Intensity*		3.06 ± 2.73 3.33; 5.42 (2.16–3.96)	1.57 ± 2.04 0.66; 2.84 (0.93–2.20)	0.01 *	0.62
*Interference*		3.14 ± 2.90 3.07; 5.33 (2.18–4.09)	1.02 ± 1.95 0.00; 1.07 (0.41–1.63)	<0.01 **	0.86
PFS (domains), mean ± SD, median; IQR, and (95% CI) ^a^					
*Behavioral/severity*		4.17 ± 2.84 3.91; 5.04 (3.23–5.10)	1.92 ± 2.46 0.53; 3.43 (1.15–2.69)	<0.01 **	0.85
*Affective*		4.68 ± 2.86 4.80; 3.90 (3.73–5.62)	2.09 ± 2.94 0.50; 3.50 (1.17–3.00)	<0.01 **	0.89
*Sensory*		4.98 ± 2.74 5.50; 3.65 (4.08–5.88)	2.10 ± 2.73 0.90; 3.25 (1.25–2.96)	<0.01 **	1.05
*Cognitive/mood*		4.57 ± 2.64 5.08; 3.75 (3.70–5.43)	1.96 ± 2.68 1.00; 2.37 (1.13–2.80)	<0.01 **	0.98
*Total fatigue*		4.61 ± 2.64 4.84; 2.85 (3.79–5.43)	2.01 ± 2.53 0.79; 2.85 (1.22–2.80)	<0.01 **	1.00
PFS (Cut-score type A), *n* (%) ^b^					
*No fatigue*	0	4 (10.5)	22 (52.4)		
*Mild*	1–3	9 (23.7)	12 (28.6)		
*Moderate*	4–6	20 (52.6)	4 (9.5)		
*Severe*	7–10	5 (13.2)	4 (9.5)	<0.01 **	-
PFS (Cut-score type B), *n* (%) ^b^					
*No fatigue*	0	4 (10.5)	22 (52.4)		
*Mild*	1–2	5 (13.2)	11 (26.2)		
*Moderate*	3–5	19 (50)	5 (11.9)		
*Severe*	6–10	10 (26.3)	4 (9.5)	<0.01 **	-

Abbreviations: LTBCSs, long-term breast cancer survivors; QLQC30, The EORTC Core Quality of Life Quality of Life Questionnaire; EVEA, Scale for Mood Assessment; IFIS, International Fitness Scale; MLTPA, Minnesota Leisure Time Physical Activity; MET, metabolic equivalent task; VAS, Visual Analogue Scale; BPI, Brief Pain Inventory; PFS, Piper Fatigue Scale; CI, confidence interval; *n*, sample size; SD standard deviation; IQR, inter-quartile range. *p* values for between-group differences were calculated using the Mann–Whitney U test for continuous variables ^a^ and *X*^2^ for categorical variables ^b^. Between-group effect sizes were calculated using Cohen’s *d* for continuous variables ^a^. * *p* < 0.05; ** *p* < 0.01.

**Table 4 cancers-17-01574-t004:** Summary of stepwise multiple linear regression analysis to determine the predictors of emotional functioning using the QLQ-C3.

Model	Variables/ Predictors	β	95% CI	*t*	*F*	*p*-Values	Regression Equation Y = a + bX
Model 1 (r^2^ = 0.573)	Role Functioning (QLQ-C30)	0.75	0.87 ± 1.30	10.23	104.65	<0.01 **	Emotional Functioning = −9.47 + (1.08 Role Functioning)
Model 2 (r^2^ = 0.601)	Role Functioning (QLQ-C30)	0.65	0.06 ± 1.18	7.68	57.88	<0.01 **	Emotional Functioning = −9.41 + (0.93 Role Functioning) + (0.18 Cognitive Functioning)
	Cognitive Functioning (QLQ-C30)	0.19	0.02 ± 0.34	2.30		0.02 *	
Model 3 (r^2^ = 0.622)	Role Functioning (QLQ-C30)	0.54	0.50 ± 1.06	5.54	41.69	<0.01 **	Emotional Functioning = −13.90 + (0.78 Role Functioning) + (−0.17 Cognitive Functioning) + (5.52 Self-Perceived General Physical Fitness)
	Cognitive Functioning (QLQ-C30)	0.18	0.01 ± 0.33	2.16		0.03 *	
	Self-Perceived General Physical Fitness (IFIS)	0.18	0.22 ± 10.81	2.07		0.04 *	
Model 4 (r^2^ = 0.642)	Role Functioning (QLQ-C30)	0.59	0.56 ± 1.13	5.97	33.57	<0.01 **	Emotional Functioning = −32.10 + (0.84 Role Functioning) + (0.24 Cognitive Functioning) + (6.15 Self-Perceived General Physical Fitness) + (2.02 Sadness–Depression)
	Cognitive Functioning (QLQ-C30)	0.26	0.07 ± 0.42	2.86		<0.01 **	
	Self-Perceived General Physical Fitness (IFIS)	0.20	0.92 ± 11.37	2.34		0.02 *	
	Sadness–Depression (EVEA)	0.18	0.03 ± 4.00	2.02		0.04 *	

Dependent variable: emotional functioning QLQ-C30; r^2^, adjusted coefficient of determination; β, regression coefficient; *t*, coefficient *t*-value; *F*, F-static of the regression model. Abbreviations: QLQC30, The EORTC Core Quality of Life Quality of Life Questionnaire; IFIS, International Fitness Scale; EVEA, Scale for Mood Assessment; CI, confidence interval. Note: For this analysis, the emotional functioning data as the dependent variable were used in their non-categorized version. *p* < 0.05 *; *p* < 0.01 **.

## Data Availability

The datasets generated during and/or analyzed during the current study are available from the corresponding author on reasonable request.

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
