# Peer review of "Emotional Functioning in Long-Term Breast Cancer Survivors: A Cross-Sectional Study on Its Influence and Key Predictors"

_cancers, 2025, doi:10.3390/cancers17091574_

Round 1

Reviewer 1 Report

Comments and Suggestions for Authors

This study aims to address the different emotional functioning between psychological distress or well-being BC survivors beyond five years post-diagnosis. The paper is very well-written with appropriate discussion and conclusion. It would be better to respond to the issues raised. 

  1. Table 1 and Discussion: It seems that a numerically higher percentage of divorced/separated people in the psychological distress group. Will that be the pre-treatment status of after? Any impact on the psychological status?
  2. Table 1 and Discussion: It is also noticed a higher percentage of having breast cancer family history in the psychological distress group. Are there any hereditary genetic data available? If not, it will be better to briefly comment/interpretate on this issue in the Discussion.
  3. Discussion (last sentence 552-553): These insights support the development of different targeted strategies/interventions aimed at enhancing both psychological "distress" and physical well-being in LTBCSs.

Author Response

Sandra Atienzar-Aroca

Department of Dentistry, Faculty of Health Sciences

European University of Valencia

46010, Valencia, Spain

Editorial Reviewer 1

Cancers

27 April 2025

Dear Reviewer 1,

This study aims to address the different emotional functioning between psychological distress or well-being BC survivors beyond five years post-diagnosis. The paper is very well-written with appropriate discussion and conclusion. It would be better to respond to the issues raised. 

Author response: First of all, we would like to thank you for your words and the time dedicated to the understanding and improvement of this scientific work. In this way, and from here on, all the answers are detailed individually to each of your suggestions or comments.

Please find below the answers to each of your contributions:

Reviewer comment: Table 1 and Discussion: It seems that a numerically higher percentage of divorced/separated people in the psychological distress group. Will that be the pre-treatment status of after? Any impact on the psychological status?

Author response:

Thank you very much for your valuable comments. As the first and second comments are related, I have addressed both questions by incorporating the following changes and adding a new paragraph to the Discussion (Lines 476 to 492):

“Although not statistically significant, the higher prevalence of family history of BC among participants with psychological distress (p = 0.06) may reflect a clinically relevant trend. The absence of genetic testing data in our sample limits further interpretation; however, it is plausible that familial cancer experiences or perceived hereditary risk could influence long-term psychological outcomes. Similarly, although marital status did not differ significantly between groups (p = 0.42), it is important to note that this variable was also assessed five years after diagnosis, and may therefore reflect both pre-existing and post-diagnosis relational dynamics. The higher proportion of divorced or separated individuals among those with psychological distress (15.8% vs 4.8%) could be relevant but must be interpreted with caution. Further research is needed to explore the directionality and potential psychological impact of both family history and marital status LTBCSs. These findings highlight the importance of not only continuing to investigate such sociodemographic and familial factors, but also of developing comprehensive psychosocial interventions aimed at enhancing emotional regulation and addressing psychological distress in LTBCSs. Implementing routine mood assessments and providing tailored psychological support could be instrumental in improving their overall well-being and HRQoL”.

Reviewer comment: Table 1 and Discussion: It is also noticed a higher percentage of having breast cancer family history in the psychological distress group. Are there any hereditary genetic data available? If not, it will be better to briefly comment/interpretate on this issue in the Discussion.

Author response:

Thank you for your valuable comment. As the issue raised regarding family history of breast cancer is closely related to the first comment on marital status and psychological distress, we have already addressed both concerns in the revised Discussion (Lines 476 to 492). In particular, we have elaborated on the potential influence of family history of breast cancer on psychological distress and the limitations due to the absence of genetic testing data in our sample, as well as the interpretation of marital status in relation to psychological distress.

We trust this adequately addresses your concerns, but please do not hesitate to let me know if further clarification is needed.

Reviewer comment: Discussion (last sentence 552-553): These insights support the development of different targeted strategies/interventions aimed at enhancing both psychological "distress" and physical well-being in LTBCSs.

Author response:

Thank you very much for this helpful suggestion. You are absolutely right, and we have now added the word distress after psychological, as recommended, to improve clarity and precision in the sentence (Lines 628 to 629).

“These insights support the development of targeted interventions aimed at enhancing both psychological distress and physical well-being in LTBCSs”.

The author responsible for correspondence is:

Sandra Atienzar-Aroca

Department of Dentistry, Faculty of Health Sciences

European University of Valencia

46010, Valencia, Spain

Sincerely,

Sandra Atienzar-Aroca

Reviewer 2 Report

Comments and Suggestions for Authors

The manuscript needs following clarifications:

  1. “Additionally, individuals with diagnosed psychiatric disorders or those taking psychotropic medication were excluded to prevent potential influences on the psychological assessments” – since it was a cross-sectional study, this should not have done. Authors should ideally select all the participants who fulfils the eligibility criteria of LTBCSs, irrespective of presence of psychiatric disease.
  2. Standardized coefficients have been correctly used in regression (Line 257). It would be better if authors mention that standardized coefficients were calculated for comparison with other independent variables.
  3. Better to mention the term “linear” regression (line 251-252). Moreover, would it be ‘multiple’ or ‘multivariable’ linear regression? Please check. Before proceeding to linear regression, please check the distribution of the both dependent and independent variables. Better to mention the dependent variable separately for the linear regression for better understanding of the readers.
  4. If Spearman’s correlation was calculated, then better not to proceed for Linear regression.
  5. Table 2: Some of the SD scores are quite near to the mean value, that indicates either presence of skewness or outlier. In that case, it would be better to use median and IQR instead of mean and SD values.
  6. It would be better to mention about the model fitness of liner regression model as running text.

Author Response

Sandra Atienzar-Aroca

Department of Dentistry, Faculty of Health Sciences

European University of Valencia

46010, Valencia, Spain

Editorial Reviewer 2

Cancers

27 April 2025

Dear Reviewer 2,

The manuscript needs following clarifications:

Author response: First of all, we would like to thank you for your words and the time dedicated to the understanding and improvement of this scientific work. In this way, and from here on, all the answers are detailed individually to each of your suggestions or comments.

Please find below the answers to each of your contributions:

Reviewer comment: “Additionally, individuals with diagnosed psychiatric disorders or those taking psychotropic medication were excluded to prevent potential influences on the psychological assessments” – since it was a cross-sectional study, this should not have done. Authors should ideally select all the participants who fulfils the eligibility criteria of LTBCSs, irrespective of presence of psychiatric disease.

Author response:

Thank you for your comment and for raising this important point. We understand your concern regarding the exclusion of individuals with diagnosed psychiatric disorders or those taking psychotropic medication in our cross-sectional study. However, we believe that this decision was justified for several reasons, which we would like to clarify.

The primary aim of our study was to assess the psychological distress of long-term breast cancer survivors in a cohort that was as homogenous as possible in terms of the factors affecting psychological well-being. Our decision to exclude individuals with diagnosed psychiatric disorders or who were taking psychotropic medications was based on the need to minimize confounding factors that could potentially distort the psychological assessments. Psychiatric disorders and psychotropic medication are known to have a significant impact on mood, cognitive function, and emotional regulation, which could have introduced bias into our measures of psychological distress. Including participants with such conditions could have resulted in an overestimation or underestimation of the psychological distress attributed specifically to the long-term effects of breast cancer, rather than due to pre-existing mental health conditions.

Moreover, while we recognize that the cross-sectional nature of the study limits the ability to infer causality, we felt it was essential to create as clear a picture as possible of the psychological state of long-term breast cancer survivors who were not influenced by other major mental health issues. This was especially important given the potential impact of psychiatric disorders on the results, which could have interfered with our primary objective of understanding the psychosocial impact of surviving breast cancer.

We also acknowledge that the inclusion of participants with psychiatric conditions could have provided valuable insights into the intersection between mental health and cancer survivorship. However, we believe this would have warranted a separate study with specific protocols and tools tailored to assessing the psychological outcomes of such individuals in a more comprehensive and controlled manner.

We hope that this explanation clarifies our reasoning behind this methodological choice. Nonetheless, we are open to further discussion and are willing to consider your suggestion if you feel it would enhance the validity of our findings.

Reviewer comment: Standardized coefficients have been correctly used in regression (Line 257). It would be better if authors mention that standardized coefficients were calculated for comparison with other independent variables.

Author response:

Thank you for your helpful comment. We appreciate your suggestion to clarify the use of standardized coefficients in our regression analysis. We have now updated the manuscript to explicitly mention that standardized coefficients were calculated to allow for a better comparison of the relative influence of the independent variables on emotional functioning. We believe this clarification improves the transparency of the statistical approach (Lines 277 to 281).

“A forward selection approach was applied, adding significant predictors in order of their association strength. At each step, model significance was assessed, and standardized β coefficients were calculated for the final model to allow for a comparison of the relative impact of the independent variables on emotional functioning”.

Reviewer comment: Better to mention the term “linear” regression (line 251-252). Moreover, would it be ‘multiple’ or ‘multivariable’ linear regression? Please check. Before proceeding to linear regression, please check the distribution of the both dependent and independent variables. Better to mention the dependent variable separately for the linear regression for better understanding of the readers.

Author response:

Thank you for your valuable feedback. We have taken your suggestions into account and made the following clarifications:

  1. We have added the term "linear" to specify the type of regression used in our analysis, as you suggested.
  2. Regarding the distribution of the variables, we agree that it is important to ensure the assumptions of regression are met. Prior to conducting the regression, we checked the distribution of both the dependent and independent variables. Given that both the dependent variable (emotional functioning) and the independent variables showed a non-normal distribution, we first performed Spearman correlation analysis to examine the relationships between the variables. This non-parametric test was appropriate due to the distribution of the data.
  1. After confirming the relationships between the variables using Spearman’s correlation, we proceeded with multiple linear regression, using the continuous dependent variable, emotional functioning, as the outcome. The regression model was constructed with independent variables that had significant correlations with the dependent variable, and we ensured that inter-variable correlations were below 0.70 to minimize collinearity. We also performed forward selection to add the most significant predictors to the model. Additionally, we verified the assumptions of linearity and homoscedasticity using residual plots.

We hope that these clarifications address your concerns regarding the analysis.

Reviewer comment: If Spearman’s correlation was calculated, then better not to proceed for Linear regression.

Author response:

Thank you for your comment and for raising an important point. We appreciate your suggestion to reconsider the use of linear regression following Spearman's correlation.

Although we acknowledge that Spearman's correlation is a non-parametric test, it was used to explore the relationships between variables, given that both the dependent and independent variables exhibited non-normal distributions. After establishing these relationships, we proceeded with multiple linear regression for the following reasons:

  1. Although our variables were not normally distributed, we confirmed the residual of the regression model were approximately normally distributed and that variance inflation factors were acceptable, supporting the validity of the linear regression approach.
  2. Multiple linear regression is a robust method that can still be applied to non-normally distributed data, provided the assumptions of the model (such as absence of multicollinearity) are met. In our case, we verified that inter-variable correlations were below 0.70 to minimize collinearity, which allowed us to proceed with the regression.
  3. We believe that multiple linear regression offers a more nuanced analysis of the relationship between emotional functioning and the independent variables, allowing us to quantify the relative impact of each predictor.

We understand that there are alternative non-parametric approaches to regression (such as ordinal logistic regression or non-parametric regression), but given the robustness of multiple linear regression and the absence of significant violations of the model's assumptions, we decided to proceed with this approach. We hope this explanation clarifies our choice of method.

Reviewer comment: Table 2: Some of the SD scores are quite near to the mean value, that indicates either presence of skewness or outlier. In that case, it would be better to use median and IQR instead of mean and SD values.

Author response:

Thank you for your insightful comment. We appreciate your suggestion regarding the use of mean and standard deviation (SD) in Table 2, particularly in the context of the potential presence of skewness or outliers.

In light of your observation, we agree that median and interquartile range (IQR) would be more appropriate measures for the variables showing such distribution characteristics. Therefore, we have revised Table 2 to replace mean and SD with median and IQR for a more accurate representation of the data, particularly given the potential skewness or outliers.

Additionally, we have made the same adjustment to Table 3 to ensure consistency across the presentation of our results.

We hope that these modifications address your concerns, and we appreciate your input in improving the clarity and robustness of our findings.

Reviewer comment: It would be better to mention about the model fitness of liner regression model as running text.

Author response:

Thank you for your valuable comment. We appreciate your suggestion to include information on the goodness of fit for the multiple linear regression model.

In response, we would like to highlight that we have already included the adjusted R² (r² adjusted = 0.642) and the p-value range (p < 0.01 to 0.04) for the final regression model in the Results section, specifically in Section 3.8 (Lines 416 to 419). These statistics demonstrate that the model explained 64.2% of the variance in emotional functioning, providing a robust indication of the model's fit.

Additionally, following your suggestion, we have now included the F-statistic for the overall model in Table 4, to provide a more comprehensive description of model fit. Consequently, and in order to improve clarity and readability, we have reformatted Table 4 into a vertical layout, instead of the previous horizontal presentation.

We hope this satisfies your request, and we appreciate your input in improving the clarity of our results.

The author responsible for correspondence is:

Sandra Atienzar-Aroca

Department of Dentistry, Faculty of Health Sciences

European University of Valencia

46010, Valencia, Spain

Sincerely,

Sandra Atienzar-Aroca

Reviewer 3 Report

Comments and Suggestions for Authors

Thank you for the opportunity to review this manuscript.
The authors demonstrated a strong correlation between emotional functioning and various aspects of health. The manuscript is interesting and provides important insights for the follow-up care of breast cancer survivors. However, the directionality of the relationship—whether emotional functioning is a cause or consequence—was not established. Further longitudinal studies are warranted to clarify this issue.

I also recommend that the authors evaluate and/or discuss the interrelationships among the different health-related scales assessed in the study, as this could provide a more comprehensive understanding of the health status of long-term breast cancer survivors.

Author Response

Sandra Atienzar-Aroca

Department of Dentistry, Faculty of Health Sciences

European University of Valencia

46010, Valencia, Spain

Editorial Reviewer 3

Cancers

27 April 2025

Dear Reviewer 3,

Thank you for the opportunity to review this manuscript.

Author response: First of all, we would like to thank you for your words and the time dedicated to the understanding and improvement of this scientific work. In this way, and from here on, all the answers are detailed individually to each of your suggestions or comments.

Please find below the answers to each of your contributions:

Reviewer comment: The authors demonstrated a strong correlation between emotional functioning and various aspects of health. The manuscript is interesting and provides important insights for the follow-up care of breast cancer survivors. However, the directionality of the relationship—whether emotional functioning is a cause or consequence—was not established. Further longitudinal studies are warranted to clarify this issue.

Author response:

We sincerely thank the reviewer for this positive evaluation and for highlighting an important conceptual aspect. We agree that the directionality of the associations reported in our study cannot be inferred. As already acknowledged in the “Strengths and Limitations” section (Lines 586 to 610), the cross-sectional nature of our design limits causal interpretation. Furthermore, we noted the absence of pre-diagnosis psychological data as a specific limitation, which prevents us from determining whether emotional distress is a consequence of survivorship or a pre-existing condition. In line with the reviewer’s suggestion, we have emphasized the need for future longitudinal research to explore the temporal dynamics and causal relationships between emotional functioning and other health-related variables in LTBCSs.

“This study has several limitations that should be acknowledged. The cross-sectional design of this study precludes causal interpretations and the use of self-reported measures may introduce recall or response bias. However, previous studies support the validity of self-reported assessments in evaluating both physical and psychological well-being, as they can effectively differentiate between individuals with varying levels of functioning [25,26]. Nonetheless, future research should consider incorporating objective assessments to complement self-reported data and provide a more comprehensive evaluation of emotional functioning in LTBCSs. Additionally, the exclusive inclusion of female LTBCSs limits the generalizability of these findings to male BC survivors, who, despite representing approximately 1% of cases, remain underrepresented in survivorship research [56]. Another limitation is the lack of pre-diagnosis psychological health data, which prevents determining whether the observed emotional distress is a consequence of cancer survivorship or an exacerbation of pre-existing psychological conditions. Longitudinal studies tracking emotional functioning from diagnosis onwards could help clarify these associations. Finally, and regarding the classification of emotional functioning, we selected a cut-off point of 90, as the findings of the previous study indicated that individuals scoring below this threshold might have unaddressed psychological needs [16]. While in another investigation the authors suggest lower cut-off values (ranging from 73.1 to 75.9) for populations with a mean age of 49 years [57] (mean age among our LTBCSs), applying these thresholds in our study could have led to the misclassification of individuals with psychological distress as having good emotional well-being. Given that LTBCSs may experience persistent but unrecognized psychological challenges, we prioritized a more conservative threshold to avoid overlooking unmet psychological needs. Future research should further investigate the appropriateness of different cut-off points in this population, considering variations in age, survivorship stage, and psychological distress presentation”.

Reviewer comment: I also recommend that the authors evaluate and/or discuss the interrelationships among the different health-related scales assessed in the study, as this could provide a more comprehensive understanding of the health status of long-term breast cancer survivors.

Author response:

Thank you for your comment. We believe we have already addressed the interrelationships between the different health-related scales throughout the discussion, particularly emphasizing how factors such as emotional functioning, health-related quality of life, self-perceived physical fitness, pain, cancer-related fatigue, and mood are deeply interconnected. Throughout the discussion, we highlight how psychological distress in long-term breast cancer survivors is associated not only with poorer health-related quality of life but also with factors like self-perceived physical fitness, mood, pain, and cancer-related fatigue. We have also pointed out how these factors interact in a complex manner, underscoring the importance of addressing both functional and psychological aspects of well-being in follow-up care.

For example, the relationship between psychological distress and physical functioning (self-perceived physical fitness and physical activity) is discussed in detail, showing how emotional distress affects patients’ ability to engage in physical activity and how improving these aspects could contribute to reducing emotional distress. Similarly, we discuss the interrelationship between psychological distress, cancer-related fatigue, and pain, and how these factors may contribute to a continuous cycle of physical and emotional discomfort.

We believe this integrated approach already provides a comprehensive understanding of the health status of long-term breast cancer survivors is and the complex interactions between the various health scales assessed. However, if this explanation is not sufficient, we would be open to the possibility of delving further into these interrelationships. Nevertheless, we believe that such an addition may increase the complexity of the manuscript, potentially making it more challenging to understand.

The author responsible for correspondence is:

Sandra Atienzar-Aroca

Department of Dentistry, Faculty of Health Sciences

European University of Valencia

46010, Valencia, Spain

Sincerely,

Sandra Atienzar-Aroca

Reviewer 4 Report

Comments and Suggestions for Authors

Your research reflects what we know about emotional functioning and the impact on health and quality of life for our cancer survivors.  What would be really important work would be moving beyond this to interventions.  Working with our clinical colleagues to develop realistic methods of evaluating these patients, when and how and what interventions work.

Author Response

Sandra Atienzar-Aroca

Department of Dentistry, Faculty of Health Sciences

European University of Valencia

46010, Valencia, Spain

Editorial Reviewer 4

Cancers

27 April 2025

Dear Reviewer 4,

Author response: First of all, we would like to thank you for your words and the time dedicated to the understanding and improvement of this scientific work. In this way, and from here on, all the answers are detailed individually to each of your suggestions or comments.

Please find below the answers to each of your contributions:

Your research reflects what we know about emotional functioning and the impact on health and quality of life for our cancer survivors.  What would be really important work would be moving beyond this to interventions.  Working with our clinical colleagues to develop realistic methods of evaluating these patients, when and how and what interventions work.

Author response: We fully agree with your observation. The objective of our current study was to explore whether the emotional functioning affectations commonly associated with cancer survivorship persist five years post-diagnosis. Our aim was to better understand the long-term impact on health and health-related quality of life, and to identify potential predictors of these outcomes.

We view this work as an essential preliminary step that can lay the groundwork for future research focused on the development and implementation of effective interventions. By identifying those most at risk and the factors contributing to long-term affectations, we hope to support clinicians in designing realistic and evidence-based evaluation methods and intervention strategies tailored to this population.

We greatly appreciate your encouragement and share your commitment to furthering research in this area with the ultimate goal of improving long-term care for cancer survivors.

Sincerely,

Sandra Atienzar-Aroca

Department of Dentistry, Faculty of Health Sciences

European University of Valencia

46010, Valencia, Spain

Round 2

Reviewer 3 Report

Comments and Suggestions for Authors

Thank you for your appropriate response.

I do not have any other comments.